# Evaluation of Immune Response to Mucosal Immunization with an Oral Probiotic-Based Vaccine in Mice: Potential for Prime-Boost Immunization against SARS-CoV-2

**DOI:** 10.3390/ijms25010215

**Published:** 2023-12-22

**Authors:** Galina Leontieva, Tatiana Gupalova, Yulia Desheva, Tatiana Kramskaya, Elena Bormotova, Irina Koroleva, Olga Kopteva, Alexander Suvorov

**Affiliations:** Scientific and Educational Center, Molecular Bases of Interaction of Microorganisms and Human of the World-Class Research Center, Center for Personalized Medicine, FSBSI, IEM, 197376 Saint Petersburg, Russia; iem@iemspb.ru (G.L.); tvgupalova@rambler.ru (T.G.); tatyana.kramskaya@gmail.com (T.K.); bormotovae@rambler.ru (E.B.); ivkoroleva@yandex.ru (I.K.); olga.s.kopteva@yandex.ru (O.K.); alexander_suvorov1@hotmail.com (A.S.)

**Keywords:** mucosal vaccines, probiotic-based vaccines, SARS-CoV-2, immune response, S protein, *Enterococcus faecium* L3, prime-boost immunization

## Abstract

Following the conclusion of the COVID-19 pandemic, the persistent genetic variability in the virus and its ongoing circulation within the global population necessitate the enhancement of existing preventive vaccines and the development of novel ones. A while back, we engineered an orally administered probiotic-based vaccine, L3-SARS, by integrating a gene fragment that encodes the spike protein S of the SARS-CoV-2 virus into the genome of the probiotic strain *E. faecium* L3, inducing the expression of viral antigen on the surface of bacteria. Previous studies demonstrated the efficacy of this vaccine candidate in providing protection against the virus in Syrian hamsters. In this present study, utilizing laboratory mice, we assess the immune response subsequent to immunization via the gastrointestinal mucosa and discuss its potential as an initial phase in a two-stage vaccination strategy. Our findings indicate that the oral administration of L3-SARS elicits an adaptive immune response in mice. Pre-immunization with L3-SARS enhances and prolongs the humoral immune response following a single subcutaneous immunization with a recombinant S-protein analogous to the S-insert of the coronavirus in *Enterococcus faecium* L3.

## 1. Introduction

The emergence of the SARS-Cov-2 pandemic and the urgent need for preventive measures to control the spread of the virus has expedited the development of antiviral vaccines [1,2,3,4,5]. Within less than a year of the outbreak of the epidemic in China, vaccines were successfully created and subjected to essential clinical trials [6,7,8,9]. Various types of vaccines are now available, including traditional inactivated vaccines (CoviVac, Sinopharm, Bharat Biotech), protein-based vaccines (EpiVacCorona, Medicage, Novavax), new vector vaccines (AstraZeneca, Sputnik V, Johnson & Johnson), and innovative RNA vaccines (Pfizer-BioNTech, Moderna) [10,11,12].

Approved vaccines have demonstrated favorable outcomes by mitigating the risks of severe illness, hospitalizations, and fatalities [13,14,15,16]. Nevertheless, empirical clinical evidence suggests that presently available vaccines and conventional vaccination schedules exhibit limited efficacy in preventing infections, and their effectiveness wanes over time [17].

The continual emergence of new sub-variants of SARS-CoV-2 capable of evading immunity keeps the high infection rates within the population (WHO. Tracking SARS-CoV-2 variants. https://www.who.int/activities/tracking-SARS-CoV-2-variants/(2020), accessed on 10 March 2023). Consequently, there is an imperative to enhance existing vaccines and develop novel formulations and vaccination schedules to address these challenges.

Numerous successful experimental models in the literature have described the utilization of bacteria as live delivery systems for vaccine antigens [18,19,20,21,22,23]. Employing bacterial vectors for antigen delivery to mucous membranes presents a rational approach, as it addresses several issues associated with parenteral vaccines. Intranasal or enteral delivery offers convenience, non-traumatic administration, and cost-effectiveness. The oral antigen delivery system elicits a robust mucosal immune response, distinguishing it from parenteral vaccination, which yields a weaker immune response at mucous membranes [18]. Moreover, integrating vaccine antigens into the structure of probiotic microorganisms confers additional advantages to such vaccines. The favorable effects of probiotic administration complement the specific immune stimulation exerted by these vaccines [24].

We previously developed and successfully employed a technique for inserting gene fragments from pathogenic streptococci into the genome of *Enterococcus faecium* L3, yielding antibacterial vaccines [22,23]. This method is specifically designed to incorporate streptococcal DNA fragments into the enterococcal gene responsible for encoding the fimbrial protein, thereby enabling the expression of foreign proteins on the surface of the probiotic bacterium. In murine models, nasal or oral mucosal administration of this bacterial vaccine elicited protective immunity.

Using the aforementioned genetic modification method of enterococcus, we prepared a probiotic vaccine targeting the SARS-CoV-2 coronavirus by integrating a gene fragment encoding the coronavirus S protein [25]. This viral antigen integrated into the structure of bacterial pili was able to evoke both innate and adaptive immune responses in experimental animals. Studies on Syrian hamsters demonstrated that recombinant vaccine strains, derived from the probiotic strain *E. faecium* L3 and expressing an immunogenic fragment of the S1 protein of the coronavirus, are both safe and capable of effectively suppressing SARS-CoV-2 infection in laboratory hamsters following oral vaccination [26].

The primary objective of this research was to evaluate the immune response specific to the SARS-CoV-2 antigen following oral administration to mice of the aforementioned vaccine strain, which we designated as L3-SARS. We analyzed the characteristics of the immune response to the antigen within the bacterial vector’s structure. Additionally, we explored the potential of utilizing the vaccine in a prime-boost approach, investigating the impact of vaccination on subsequent single parenteral administration of a homologous antigen.

## 2. Results

We investigated the immunogenicity of the previously obtained probiotic strain *E. faecium* L3, which was engineered to carry the coronavirus spike S protein, through oral immunization in mice.

### 2.1. Study of the Persistence of Modified Enterococcus in the Gastrointestinal Tract of Mice

We employed the vaccination regimen previously used in our studies of vaccines against pathogenic streptococci [22,23]. The vaccination procedure consisted of a single oral administration of the drug for three consecutive days. This procedure was repeated three times at intervals of two weeks. Thus, mice received the vaccine orally on days 1–3, 14–16, and 28–30 from the start of the experiment. To examine the persistence of the vaccine strain in the gastrointestinal tract of mice, we directly detected the modified enterococcal strain using selective azide agar. Azide (enterococcal) agar inhibits the growth of Gram-negative bacteria, allowing the growth of enterococci that form pink colonies on the agar surface (Appendix A). Erythromycin was incorporated into the agar to facilitate the detection and enumeration of erythromycin-resistant L3-SARS colonies.

As anticipated, in the presence of erythromycin, only the modified strain proliferated and produced characteristic pink colonies on the enterococcal agar (Appendix A). Evaluation of the intestinal microbiota of intact mice revealed the presence of enterococci in fecal samples, but they did not grow on azide agar supplemented with erythromycin (Appendix A). Thus, by plating fecal sample homogenates on the agar, we successfully determined the duration of persistence of the modified strain in the gastrointestinal tract of vaccinated mice. Additionally, to validate the results, we isolated DNA from typical bacterial colonies and employed it as a template in PCR with primers specific to the coronavirus insert (Appendix A). Primers K1 and K2 flank the region of the viral insert, while primers B1 and K2 flank the region between enterococcal DNA and the viral insert. The presence of the corresponding PCR products confirms the presence of a viral insert in the enterococcus genome.

Analysis of the microbiota in the feces of vaccinated mice after the last oral vaccination revealed that the modified enterococcus persisted in the gastrointestinal tract and could be isolated from the feces up to 9 days after vaccination (Figure 1).

The first fecal sample was taken on day 4 from the start of vaccination (day 1 after the third and last doses of the vaccine, respectively).

### 2.2. Study of the Immunogenicity of the Coronavirus Protein S in the Composition of L3-SARS

Balb/c mice were orally immunized according to the schedule described in the Materials and Methods section. Blood serum and nasal washings were collected from mice after each immunization and two weeks following the completion of the immunization protocol. ELISA analysis of the samples revealed the presence of S-specific IgA and IgG antibodies in the blood serum, as well as IgA in nasal lavages, following oral immunization with the experimental vaccine L3-SARS (Figure 2).

Specific serum IgA accumulated prominently after vaccination in Balb/c mice. At 28 days from the initiation of oral vaccination, the serum IgA antibody content was statistically higher than the control values, and these higher levels persisted until day 42 (Figure 2B).

Secretory anti-S IgA was detected in nasal lavages on day 40 (Figure 2C).

Further clarification is required for these specific IgG analysis data presented in Figure 2A. 

The presence of statistically significant differences in IgA responses between the L3 and L3-SARS groups was evident when serum was subjected to ELISA without additional treatment. However, such distinctions were not observed in the analysis of specific IgG (Figure 3). 

Examination of OD450 values at each data point within the average specific IgG titration curves indicated differences between the sera of the control and experimental groups (Figure 3A). Nevertheless, it was imperative to establish the statistical significance of these differences and calculate the final titer based on these obtained data. Subsequent processing facilitated the achievement of this result. Notably, the differences became more pronounced following the preliminary co-cultivation of sera with *Enterococcus faecium* L3. Consequently, the titration curves of control sera decreased while the immune curves increased (Figure 3B).

Following the aforementioned treatment, it could be concluded that specific immunoglobulins G accumulated in the serum, leading to a significant difference in the geometric mean serum titers of vaccinated mice compared with controls after 42 days, as depicted in Figure 2A.

In summary, these obtained data indicate that the 9-dose course (three times repeated three-day course) of oral vaccination stimulated the humoral and secretory immune response of mice to the S protein fragment expressed by the probiotic vaccine strain.

A comparative evaluation of control and immune mice demonstrated the induction of specific IFN-secreting T-cells in spleen lymphocytes after the course of enteral vaccination with L3-SARS.

Splenocytes from mice fed with the vaccine were stimulated in vitro with S antigen for three and six days, as described in the Materials and Methods section. Evaluation of IFN-γ response from mouse splenocytes stimulated by recombinant S antigen revealed increased IFN-γ secretion in the L3-SARS group compared to the controls, which consisted of unmodified *Enterococcus faecium* L3 or non-vaccinated mice (Figure 4).

### 2.3. Probiotic Vaccine in the Prime-Boost Method: Investigating the Effect of Vaccination on Subsequent Single Parenteral Administration of a Homologous Antigen

The unexpected emergence of the SARS-CoV-2 pandemic necessitated rapid and unprecedented protective measures. Developing an optimal immunization strategy takes time. Considering the interest in selecting the most effective vaccination regimens, we investigated the priming effect of oral vaccination with a probiotic vaccine on subsequent single subcutaneous administration of the S protein.

For this purpose, mice were immunized according to the schedule described in the Materials and Methods section. We assessed the humoral IgG immune response magnitude after a single subcutaneous S protein administration in untreated and those previously treated with L3-SARS vaccine strains and unmodified probiotic *E. faecium* L3 strain (Figure 5) on days 28 and 165 since the experiment’s inception. 

Comparison of serum IgG levels in ELISA demonstrates that priming with the probiotic vaccine enhances and prolongs the circulation of antigen-specific IgG in the bloodstream. Importantly, the time interval between the last administration of the probiotic vaccine and the booster subcutaneous immunization does not impact the overall pattern. The humoral immune response of primed mice progressively increases until the end of the observation period (days 70 and 76). In contrast, a single subcutaneous immunization of intact mice induces a noticeable immune response by days 14 and 18, which remains relatively unchanged for two weeks before declining (Figure 5A,C).

Throughout the observation period, a more noticeable accumulation of specific IgA was observed in mice that received the original and modified probiotic variants before a single subcutaneous injection of recombinant S protein, compared with the intact control. However, this excess can be considered as a trend, and the differences were not statistically significant (Figure 5B).

Clearly, preliminary oral immunization with the L3-SARS probiotic strain enhances the immune reaction to the one-time subcutaneous injection of S-protein. Thus, oral administration of the modified probiotic strain with the insertion of the viral protein gene creates a favorable immunological background for enhancing the immune response to the subsequent single subcutaneous injection of the recombinant protein encoded by this gene.

## 3. Discussion

The unexpected emergence of the coronavirus pandemic has led to the rapid development, clinical trials, and approval of various vaccines, including novel technologies [27]. These vaccines encompass live attenuated vaccines, inactivated whole virus vaccines, protein-based vaccines, viral vectors, and nucleic acid-based vaccines [10,28]. Another promising approach is the use of probiotic-based vaccines, where beneficial bacteria serve as vectors to deliver viral antigens.

Over the past two decades, this technology has been utilized to develop mucosal vaccines targeting respiratory and non-respiratory pathogens [19,22,23,29,30,31]. Lactic acid bacteria strains have been genetically modified to produce a variety of antigens. While many projects for mucosal vaccines are in preclinical stages [31,32], some, such as vaccines against human papillomavirus, have progressed to clinical trials [33,34,35]. Notably, a laboratory variant of a mucosal vaccine targeting a coronavirus, the transmissible gastroenteritis virus, causing fatal infections in pigs, was based on genetically modified probiotics [36].

Currently, Symvivo Corp. (Melbourne, Victoria, Australia) is conducting the first human trial of a probiotic-based oral vaccine against SARS-CoV-2 (bacTRL-Spike-1). The vaccine employs recombinant *Bifidobacterium longum* to deliver plasmids carrying the full-length S-protein gene [28]. The study was completed in August 2022 (https://clinicaltrials.gov/study/NCT04334980?cond=COVID-19&term=bacTRL-Spike-1&rank=1, accessed on 30 June 2023), and the results are pending.

In this study, we analyzed the immunological properties of our experimental bacterial vector coronaviral vaccine. We inserted a DNA fragment encoding a portion of the S1 protein of SARS-CoV-2 into the genome of *Enterococcus faecium* L3. This inserted fragment is transcribed, leading to the synthesis of the coronavirus protein on corresponding messenger RNAs, detectable on the surface of L3-SARS [25]. It is noteworthy that we have previously successfully employed this genetic engineering technique to obtain vaccine variants targeting GBS, pneumococci [22,23], and, more recently, influenza A virus [37,38].

The constructed vaccine strain L3-SARS was utilized for oral vaccination of mice following a schedule consisting of three cycles of administration with a two-week interval. Each cycle included three daily administrations. Literature sources suggest that four [39] to thirty [40] oral immunizations may be required to induce a local and systemic immune response to oral vaccination with modified lactic acid bacteria.

The use of probiotic microorganisms as vectors enables the long-term persistence of the vaccine antigen due to bacterial multiplication in the gastrointestinal tract. We demonstrated that after the last oral administrations, the SARS-CoV-2 vaccine strain can be detected in mice feces for up to 9 days. However, further investigation is needed to determine how the frequency of vaccination affects the persistence of modified enterococcus.

In our experimental setup, we detected IgA and IgG-specific antibodies in the serum of immunized mice after the second immunization cycle, and these antibodies continued to accumulate after the third cycle (Figure 2A,B). Additionally, after the third vaccination cycle, S-specific secretory antibodies were found in the pool of lavages from the nasopharynx and oropharynx (Figure 2C).

Analyzing the IgG humoral immune response to the recombinant vaccine revealed some peculiarities. Comparing the ELISA titration curves, we observed that the staining intensity of serial dilutions of immune sera was statistically higher than in the control group, indicating the accumulation of specific antibodies in the serum of immune animals (Figure 3A). However, these data could not be processed to calculate the end-point titer. We noticed that sera from the control animals treated with the original variant of *E. faecium* L3 contained IgG capable of nonspecific reactions with the S protein at the bottom of the plate. To address this, we assumed that such molecules could be removed by nonspecific adsorption. Indeed, after adsorption of sera on the original *E. faecium* L3, the OD 450 values in the control group decreased. It appears that oral immunization with a probiotic vaccine not only stimulates the synthesis of S-specific IgG but also leads to the accumulation of poly-specific IgG in circulation. During ELISA, these antibodies bound to the protein at the bottom of the plate, increasing the intensity of the color reaction in the control samples. Zeng MY et al. previously reported T-independent induction of poly-specific IgG antibodies by microbiota in mice [41].

Interestingly, a similar treatment of immune sera led to an increase in OD450 values. It is possible that other proteins capable of binding to the S protein at the bottom of the plate and preventing interaction with specific IgG of immune sera could be adsorbed on enterococcus. Some of these proteins may belong to mouse serum IgA.

Approximately 80% of all antibody-secreting cells in mammals reside in the gut and express the IgA isotype [42,43,44]. It has been shown that a constant process of immune response to the microbiota develops in the intestines of mammals to maintain bacterial homeostasis [45]. The pool of IgA antibodies contains both highly specific immunoglobulins A [46] and low-affinity poly-specific antibodies [47,48].

Within the intestine, two humoral immunity mechanisms coexist. The first involves a homeostatic response to commensal microorganisms, developing with limited T cell assistance, yielding antibodies with low somatic mutation levels, low affinity, and broad specificity. The second, a protective response to pathogens, generates high-affinity, specific antibodies in germinal centers, mirroring systemic responses [44].

Genetically modified probiotics, hybrids of commensal and pathogenic microorganisms, induce an immune response reflecting their dual nature. Notably, coronavirus epitope-specific antibody production within the bacterium occurs simultaneously with the immune reaction to *Enterococcus faecium* L3 itself. Evaluating the humoral immune response in the control group’s sera, subjected to oral exposure to the unmodified enterococcus strain, revealed increased IgG antibodies capable of non-specifically binding to protein S, compared to untreated mouse control group sera (Appendix A).

The role of poly-specific low-affinity antibodies induced by an orally administered probiotic vaccine should not be underestimated. Such antibodies have been shown to play a crucial role in protection against pathogenic bacteria [48,49] and viruses [50]. They also participate in the activation of the complement through an alternative pathway [51] and in the induction of cytokines involved in innate immune responses [52]. Therefore, the immune response to a probiotic vaccine not only stimulates the mechanisms of innate and adaptive defense but also their interplay.

Following the depletion of sera by adsorption on *Enterococcus faecium* L3 and the subsequent decrease in nonspecific interactions, we could compare the level of S-specific antibodies in the control and immune sera (Figure 3). These data indicate that the fragment of the coronavirus S protein, administered orally as part of a recombinant *E. faecium* strain, stimulated the development of a specific local and systemic humoral immune response. These findings complement the results of other studies on the development of local and systemic humoral immune responses to recombinant lactic acid bacteria carrying genes of pathogenic bacteria and viruses [32].

We have observed that oral immunization with a probiotic vaccine stimulated the accumulation of specific IgA in the serum (Figure 4). Despite extensive knowledge about the role of IgG in the immune defense system, the significance of serum IgA remains insufficiently studied. Previous research has shown that IgA induction occurs in lymphoid tissues of mucous membranes, particularly in the intestine. Intestinal secretory IgA and serum IgA have been found to be clonally related [53]. Notably, mucosal IgA has been shown to play a more critical role in protection against viral infections than IgG in certain cases [54].

Specific virus-specific secretory IgA has been detected in the milk of mothers infected with SARS-CoV-2, indicating that sIgA is produced in response to natural infection and transferred to newborns for protection [55]. Moreover, early humoral responses specific to SARS-CoV-2 have been found to be dominated by IgA, significantly contributing to virus neutralization. The absence of anti-SARS-CoV-2 serum IgA and secretory IgA may explain the severity of COVID-19, vaccine ineffectiveness, and prolonged viral shedding in patients with primary antibody deficiencies, including selective IgA deficiency [56].

Considering these findings, the high level of IgA following vaccination with the recombinant probiotic vaccine L3-SARS may significantly contribute to its protective efficacy.

Our data clearly demonstrate that oral vaccination with L3-SARS not only induces a humoral immune response but also stimulates the cellular immune response. In vitro, stimulation of splenocytes with the recombinant S protein resulted in a significant increase in interferon-gamma production in immune splenocytes compared to the control groups (see Figure 4). It is well-established that effector T cells CD4 Th1 and CD8 CTL produce IFN-γ in response to antigenic stimuli after the development of an adaptive immune response. Similar development of a cytotoxic immune response in mice following oral vaccination with various antiviral probiotic vaccines has been demonstrated in other studies [30].

The capability of recombinant probiotics to stimulate T-cell-mediated immune responses, in addition to IgG and sIgA production, broadens their potential and offers new opportunities for protection against viral infections [57,58,59], including the current SARS-CoV-2 infection [26]. 

In light of the COVID-19 outbreak, the pursuit of optimal vaccine regimens to enhance and maintain the protective immune response in the population is ongoing. Prime-boost vaccination strategies are being extensively investigated [60,61]. Experimental models have indicated that a prime-boost approach combining different vaccine platforms may be more effective than a single vaccine for protection against infectious diseases. Such strategies have the potential to enhance both cellular and humoral immunity [57].

To explore the results of vaccination, we followed an approach of administering the oral probiotic L3-SARS vaccine followed by a single parenteral administration of recombinant S protein.

To assess the efficacy of combining oral probiotic vaccines with parenteral protein vaccines, we employed an approach involving oral L3-SARS probiotic vaccine administration followed by a single parenteral recombinant S protein dose.

It is known that a single parenteral administration of recombinant protein vaccines is insufficient to elicit a long-lasting and robust humoral immune response [58]. Typically, vaccination protocols include at least two doses of vaccinations. Replacing one subcutaneous dose with a course of oral administration of a probiotic vaccine could be advantageous. In our study, we compared the development of the humoral immune response to a single injection of S protein in intact mice with that in mice previously immunized through the enteral route. Booster vaccinations were administered at short (14 days) and long (150 days) intervals after completing the probiotic vaccine immunization.

In both scenarios, similar trends were observed. Following a single subcutaneous immunization of mice with the S protein, the level of specific IgG antibodies increased by days 14–18 post-administration, remained relatively stable for a period, and then declined. Conversely, in mice previously vaccinated with the probiotic vaccine, a gradual and sustained accumulation of statistically higher levels of IgG antibodies was observed, persisting until 70–76 days after the subcutaneous immunization with the S protein (Figure 5A,C) and continued throughout the observation period.

Oral vaccination with the probiotic vaccine establishes a conducive immunological background, prolonging the IgG humoral response to a single parenteral administration of recombinant S protein, akin to the antigenic insertion into enterococcus. Significantly, the kinetics of specific IgG accumulation in blood post subcutaneous boosting did not fully align with the classical secondary immune response pattern seen in parenteral vaccination, marked by a swift rise in specific antibody concentration compared to the primary immune response.

Although we achieved an enhancement of the immune response’s strength and duration, the acceleration of the immune response was not observed. The observed delay in the secondary immune response could be attributed to the oral route employed during the initial stage of immunization. In this context, memory T cells are not recirculated in the bloodstream; instead, they are primarily concentrated in the mucous membranes as tissue-resident memory cells (TRM) [62,63,64]. Upon subcutaneous immunization of these mice, memory T cells are activated, albeit with some delay, as indicated by the prolonged accumulation of specific IgG in the blood serum compared to a single parenteral immunization. Notably, the immunological memory in mice persists for at least five months after the final administration of the probiotic vaccine. Further research is warranted to investigate this immunological phenomenon.

Regarding the IgA immune response in the short pre-boost scenario, significant differences in the level of S-specific antibodies were observed on the 14th day after subcutaneous administration. However, the accumulations of IgA capable of binding protein S at the bottom of the plate were evident not only in the L3-SARS-primed group but also in mice that received the unmodified version of Enterococcus. Subsequently, after 28 days, the level of specific IgA in the enterococcus-primed mice decreased (Figure 5B).

We are currently investigating the protective efficacy of the constructed vaccine strain L3-SARS. Similar live recombinant influenza vaccines modified with the NA and HA inserts of the influenza virus have already demonstrated a protective effect [37,38]. In conclusion, our results suggest that the *Enterococcus faecium* L3 strain containing epitopes of SARS-CoV-2 S protein can stimulate a specific immune response against SARS-CoV-2. Oral 9-dose vaccination of laboratory mice leads to the development of a humoral and cellular S-specific immune response. The use of a probiotic vaccine in the initial stage of vaccination enhances the immunogenic effect of subsequent single parenteral administration of the recombinant S protein.

The research on the vaccine is ongoing, and in case of positive results, L3-SARS can be considered for independent use as a prophylactic vaccine or in combination with parenteral recombinant vaccines to reduce their administration frequency.

## 4. Materials and Methods

### 4.1. Bacterial Cultures

The original *E. faecium* L3 strain and the genetically modified L3-SARS strain were cultivated in sterile THB with 0.5% yeast extract, followed by incubation at 37 °C for 24 h. Subsequently, the bacteria underwent three rounds of centrifugation at 3500 rpm for 20 min to ensure thorough washing. The resulting bacterial sediment was then suspended in PBS to achieve the desired concentration. This suspension was utilized for mouse vaccination.

For the cultivation, quantification, and identification of *E. faecium* L3 and erythromycin-resistant enterococcal transformants, LB agar (Lennox L agar, Sigma-Aldrich, St. Louis, MO, USA) and Enterococcus Differential Agar Base (TITG Agar Base) (Himedia, Mumbai, Maharashtra, India) were employed as solid media. LB agar was used without antibiotics, while agar containing 5 μg/mL of erythromycin was utilized to identify erythromycin-resistant strains.

### 4.2. Animal Procedures

Female inbred Balb/c mice, aged 10 weeks, were procured from the laboratory animal nursery “Rappolovo” in the Leningrad Region, Russia, and were employed in the experiments. The mice were housed under standard laboratory conditions, with access to food and water ad libitum.

The experimental procedures adhered to the principles outlined in the EU Directive 2010/63/EU for animal experiments and were approved and conducted in accordance with the guidelines and under the supervision of the local Biomedical Ethics Committee. The ethics committee meeting held on 28 January 2021 documented the approval of these experiments (Minutes of the meeting 1/21).

### 4.3. Immunization Schemes

Two immunization schemes were implemented to assess the immunogenic properties of the *E. faecium* L3-SARS vaccine strain through enteral modes of administration. Mice were randomly divided into two groups, each containing 10 mice (Figure 6). A total volume of 500 μL, comprising *E. faecium* L3-SARS-CoV-2 or *E. faecium* L3 at a dose of 1 *×* 109 CFU per mouse, was administered via a mouse feeding needle on days 1–3, 14–16, and 28–30 from the commencement of the experiment. Blood samples were collected 12 days after the last dose and stored at −20 °C. Nasopharyngeal and oropharyngeal lavages were pooled two days before this collection. The lavages were obtained shortly after intraperitoneal injection of 0.5% Pilocarpine hydrochloride (FARMAK, Kyiv, Ukraine) when the mice started producing increased saliva and nasal secretions. The first 50 μL of secretions were used for IgA analysis. In the manuscript, these samples are referred to as nasal lavages.

The mice were divided into two groups: one group received *E. faecium* L3-SARS, and the other received *E. faecium* L3 at an equal dose. Vaccination was administered orally on days 1–3, 14–16, and 28–30 from the commencement of the experiment. Nasopharyngeal and oropharyngeal lavages were combined on day 40, and blood samples were collected on day 42 after the initial vaccine dose.

The prime-boost effect of enteral vaccination was evaluated according to the following schedule (Figure 7). Mice were divided into three groups, each containing 20 mice. Two groups received 500 μL of *E. faecium* L3-SARS or *E. faecium* L3 at a dose of 1 × 109 CFU per mouse on days 1–3 and 14–16 from the beginning of the experiment. The mice in the third group remained intact. On the 28th day, control intact mice and mice from both experimental groups (10 mice/group) were subcutaneously injected with 20 μg of recombinant S protein in 0.2 mL. Blood samples were collected on days 0, 5, 14, 28, and 70 after the subcutaneous injection and stored at −20 °C. On day 165, rest control intact mice and mice from both experimental groups (10 mice/group) were subcutaneously injected with 20 μg of recombinant S protein in 0.2 mL. Blood samples were collected on days 0, 18, 48, and 76 after the subcutaneous injection and stored at −20 °C. 

The mice were divided into three groups. Two groups received *E. faecium* L3-SARS or *E. faecium* L3 at equal doses. The mice in the third group remained intact. On the 28th day, control intact mice and mice from both experimental groups were subcutaneously injected with recombinant S protein. Blood samples were collected on days 0, 5, 14, 28, and 70 after the subcutaneous injection. On day 165, rest control intact mice and mice from both experimental groups were subcutaneously injected with recombinant S protein. Blood samples were collected on days 0, 18, 48, and 76 after the subcutaneous injection.

### 4.4. Enzyme-Linked Immunosorbent Assay (ELISA) 

The ELISA assay was conducted following the previously described procedure [22]. Maxisorb 96-well plates (Nunc; Roskilde, Denmark) were coated with 0.25 μg/mL of S protein in 0.1 M sodium carbonate buffer, pH 9.3, overnight at 4 °C. A series of twofold dilutions of the samples (100 μL) were added to duplicate wells and incubated for 1 h at 37 °C. Between each step, the plates were washed with a blocking buffer (0.05% Tween-20 in PBS). The same buffer was used for serum and reagent dilution. HRP-labeled goat anti-mouse IgA or IgG antibodies (Sigma-Aldrich, St. Louis, MO, USA) were added (100 μL/well). After incubation at 37 °C for 1 h, the plates were developed with 100 μL/well of TMB substrate (BD Bioscience, Drachten, The Netherlands). Color development was detected after 20 min of incubation, and the reaction was stopped with 30 µL of 50% sulfuric acid. The endpoint ELISA titers were expressed as the highest dilution that yielded an optical density at 450 nm (OD450) greater than the mean OD450 plus 3 standard deviations of negative control wells.

In some cases, we constructed line graphs of serum titration and compared the average OD450 values for each data point on the graph.

### 4.5. Serum Processing for Specific IgG Analysis 

To eliminate nonspecific binding, control, and immune serum samples were diluted 200 times, mixed with *Enterococcus faecium* L3 to a final concentration of 5 × 108 CFU/mL, and incubated at 37 °C for 1 h. The bacteria were then precipitated by centrifugation for 10 min at 3500 rpm, and the precipitate was discarded. Subsequently, the serum samples were analyzed using the ELISA method.

### 4.6. Evaluation of IFN-γ Production 

To assess the cellular immune responses induced by L3-SARS, we stimulated mouse splenocytes from the L3-SARS, L3, and non-vaccinated groups with purified recombinant S protein at a final concentration of 1 μg/mL. Spleens were gently homogenized in glass homogenizers while maintaining cold conditions, and large tissue fragments were removed by filtration through Corning Cell Stainer (Sigma Aldrich, St. Louis, MO, USA). Spleen cells were cultured in triplicate wells at a concentration of 2 × 10^6^/mL in 1 mL of IMDM medium (Biolot, St. Petersburg, Russia) supplemented with 10% calf serum (BioWest, Nuaillé, France). After 3 and 6 days, cell culture supernatants were collected for interferon-gamma (IFN-γ) analysis using commercially available cytokine quantitative ELISA kits (R&D Systems, Minneapolis, MN, USA), following the manufacturer’s protocols. Cell viability was assessed using the trypan blue dye exclusion method (Sigma Aldrich, St. Louis, MO, USA), as per the manufacturer’s instructions. LPS served as a positive control, while the culture medium alone served as the negative control.

### 4.7. Evaluation of L3-SARS in Mice Feces 

The concentration of L3-SARS was determined by plating fecal homogenates on the surface of selective agar. Enterococcal (azide) agar containing erythromycin at a concentration of 5 μg/mL was utilized as a selective medium. Feces from mice were collected daily for 12 days after the third stage of vaccination and frozen at −80 °C. Fecal samples were homogenized in a Retsch MM 400 vibratory mill (Haan, Germany) at 30/s for 30 s. Subsequently, 100 μg of fecal samples were homogenized in 1 mL of PBS. Then, 10 μL of the supernatant was applied to the agar surface, and the plates were incubated for 24 h at 37 °C for colony counting. DNA from the Enterococcal colonies was tested in a PCR to confirm viral insertion. The following primers were employed: K1: TTGCATATGGGTTTCCAACCCACT (Forward)K2: GTAGAATTCGTTGTTGACATGTTCA (Reverse) B1: TGAGTGAACCACAGCCAGAA (Forward)

### 4.8. Statistical Analyses 

The results are presented as the mean ± SEM. Data were analyzed using the statistical module of GraphPad Prism 6 software (GraphPad Software, Inc., San Diego, CA, USA). Statistically significant differences between groups were determined by ANOVA with Tukey’s multiple comparison test. *p* values of <0.05 were considered significant. 

## Figures and Tables

**Figure 1 ijms-25-00215-f001:**
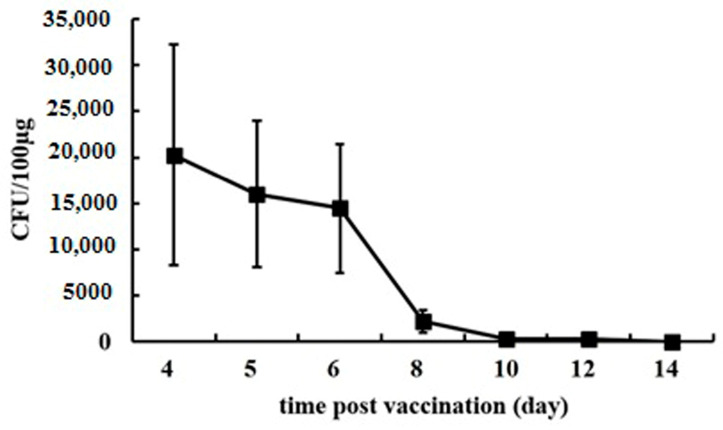
Assessing L3-SARS persistence in vaccinated mice’s gastrointestinal tract. Fecal samples (100 μg) were homogenized in 1 mL PBS, followed by 2 min of settling to remove larger particles. Afterward, 10 μL of supernatant was aseptically plated on agar and incubated at 37 °C for 24 h to enumerate colonies.

**Figure 2 ijms-25-00215-f002:**
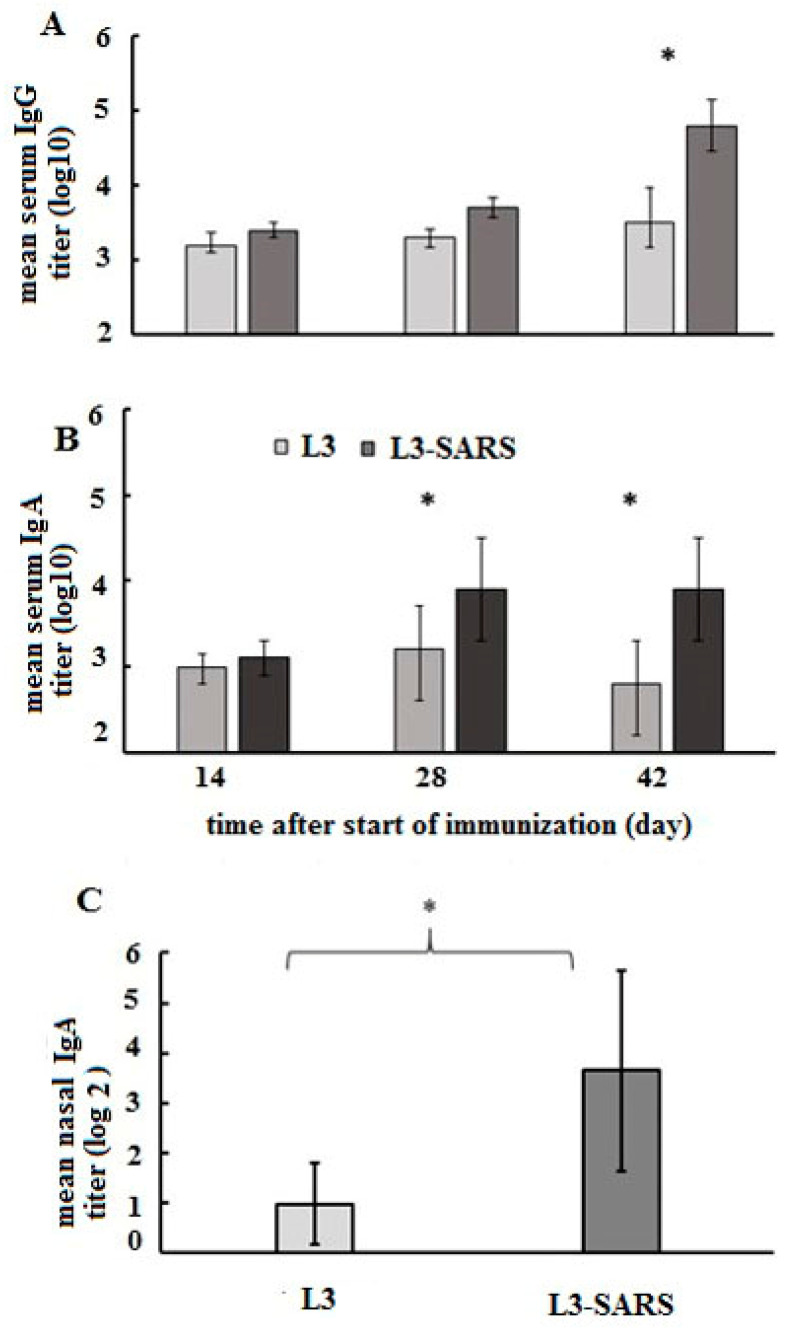
S-specific serum and local immune response after oral immunization with enterococcal strains. Serum and nasal lavages were collected from mice (*n* = 10) on the indicated days in the figure. Nasal lavages were collected on day 40 after the start of immunization. (**A**) Sera were treated with *Enterococcus faecium* L3 enterococci, and the level of IgG antibodies against recombinant protein S was assessed using ELISA. (**B**) The level of serum IgA antibodies against recombinant protein S was assessed using ELISA. (**C**) In nasal lavages, the level of IgA antibodies was assessed using ELISA. Reciprocal antibody titers were expressed as log10 and presented as mean ± SEM on the ordinate axis, (*)*—p* ≤ 0.05.

**Figure 3 ijms-25-00215-f003:**
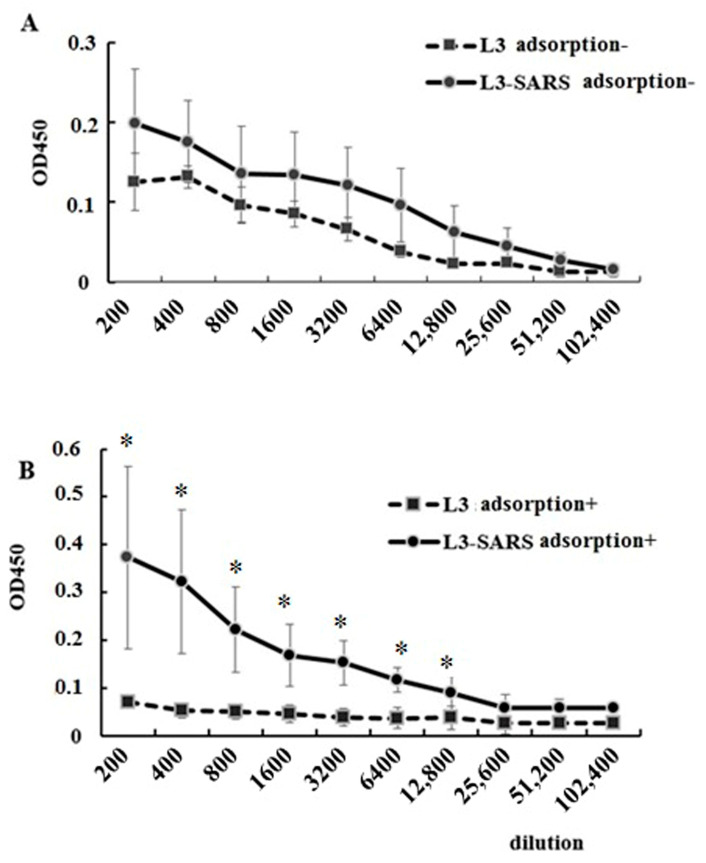
Serum titration curves comparing before and after treatment with *Enterococcus faecium* L3. (**A**) Titration curves of native control (L3) and immune (L3-SARS) sera. (**B**) Titration curves of control (L3) and immune (L3-SARS) sera after treatment with *Enterococcus faecium* L3. To remove nonspecific IgG binding, control and immune sera were incubated with *Enterococcus faecium* L3 as described in the Materials and Methods section. Graph lines show mean OD450 values for each serum dilution and are presented as mean ± SEM on the ordinate axis, (*)*—p* ≤ 0.05.

**Figure 4 ijms-25-00215-f004:**
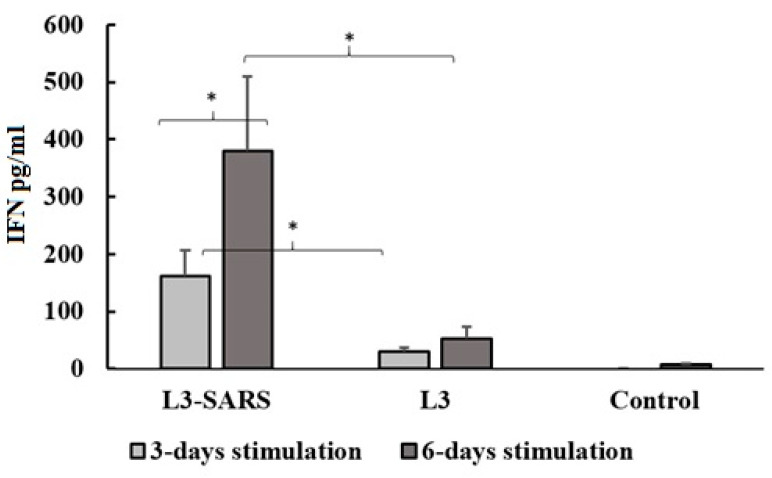
Induction of IFN-γ in spleen lymphocytes after S antigen stimulation in vitro. Mouse splenocytes were stimulated with purified recombinant S protein at a final concentration of 1 μg/mL. After 3 and 6 days, cell culture supernatants were collected to examine the amounts of interferon-gamma (IFN-γ) using commercially available cytokine quantitative ELISA kits. IFN γ concentration is presented as mean ± SEM after subtracting the values of non-stimulated splenocytes obtained from the same groups of mice, (*)*—p* ≤ 0.05.

**Figure 5 ijms-25-00215-f005:**
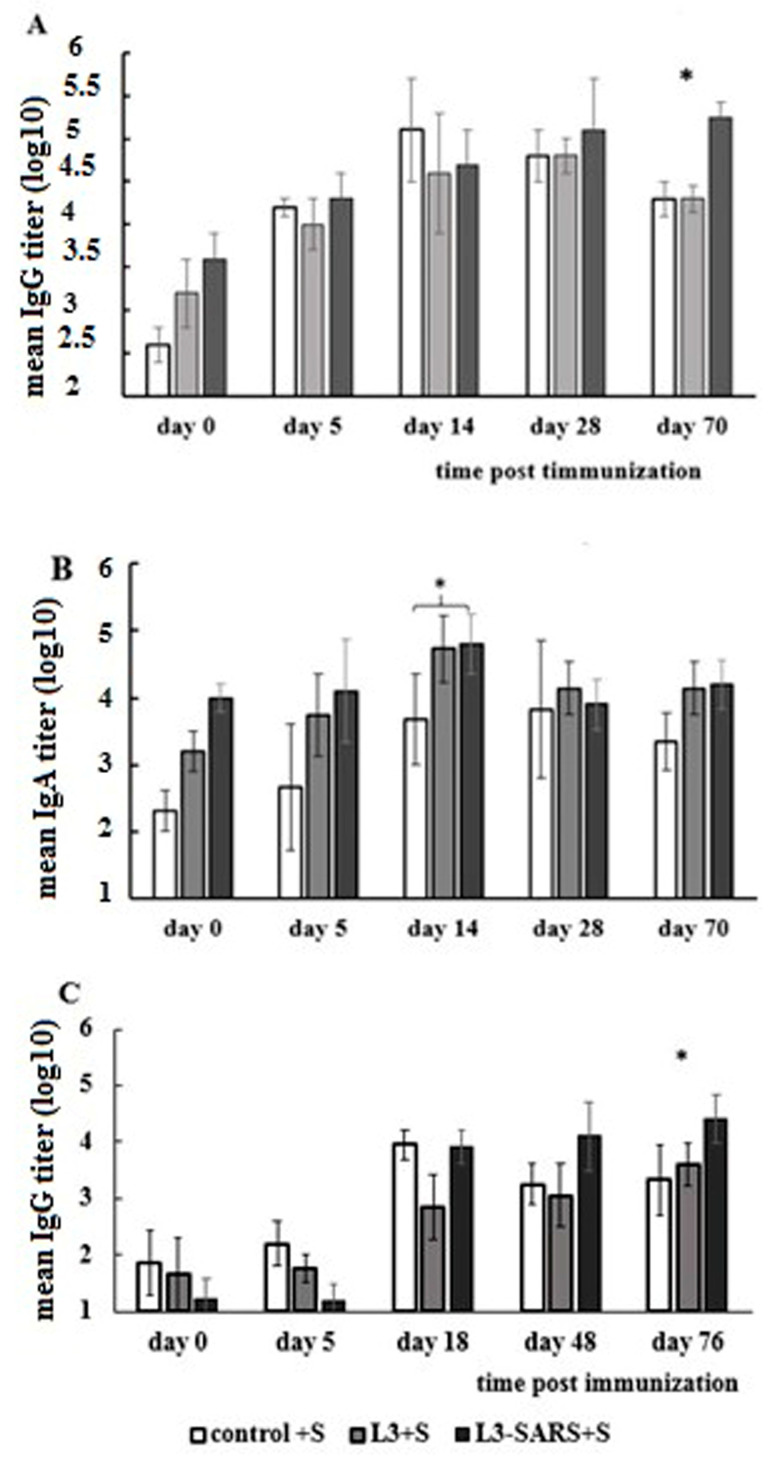
S-specific serum immune response to heterologous prime-boost with initial oral L3-SARS and subsequent subcutaneous protein S immunization. At 12 day (**A**,**B**) and 150 day (**C**) intervals, prime-boost vaccination. Levels of IgG (**A**,**C**) and IgA (**B**) antibodies against recombinant protein S were assessed using ELISA. Reciprocal antibody titers were expressed as log10 and presented as mean ± SEM, (*)*—p* ≤ 0.05.

**Figure 6 ijms-25-00215-f006:**
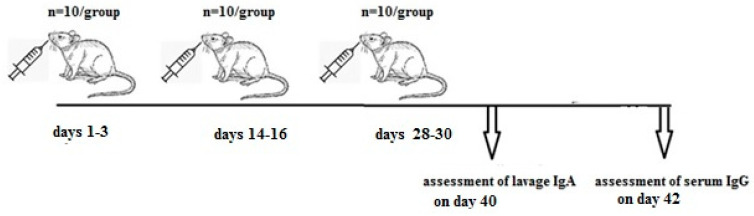
The schedule of the vaccination.

**Figure 7 ijms-25-00215-f007:**
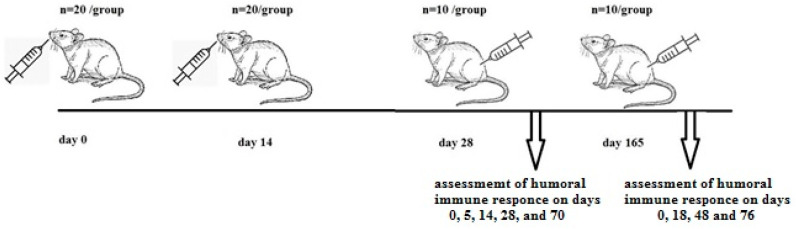
The schedule of the prime-boost effect of enteral vaccination.

## Data Availability

Data are contained within the article and Appendix A. The data that support the findings of this study are available on request from the corresponding author A.S.

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
