# Peer review of "Evaluation of Immune Response to Mucosal Immunization with an Oral Probiotic-Based Vaccine in Mice: Potential for Prime-Boost Immunization against SARS-CoV-2"

_ijms, 2023, doi:10.3390/ijms25010215_

Round 1

Reviewer 1 Report

Comments and Suggestions for Authors

Comments on the Quality of English Language

Author Response

We express our gratitude to all the reviewers for their thoughtful and constructive evaluation of our article. Their critical insights and kind considerations contribute significantly to enhancing the quality of our work and positively influence our understanding of the subject under study.

If space is allowed in this journal, I would have them include most of the supplementary information in the manuscript itself. At minimum, include S5 showing the design and group numbers.

Reply. We thank the reviewer for this remark. We included the schedules of both experiments in the Materials and Methods section.

The authors address some of the challenges they encountered with respect to nonspecific reactivity from L3 alone and potential reasons (which is interesting) – it would be great to follow up with experiments to look at local versus systemic memory T cells, and also what degree of protection (if any) is conferred by L3 alone compared with L3-SARS. These would improve the manuscript but are not necessary for publication.

Reply. We fully endorse the reviewer’s perspective on the significance of investigating the contribution of nonspecific protection to the overall efficacy of the probiotic oral coronavirus vaccine. This aspect necessitates dedicated exploration. Currently, we are undertaking such investigations using a model involving two vaccine strains of E. faecium L3 that incorporate antigens from the influenza A virus and S. pneumoniae. This model proves advantageous as it not only facilitates the study of the immune response but also allows for the assessment of the protective efficacy of vaccine formulations by comparing the contributions of the probiotic strain and the incorporated vaccine antigens. It is lot harder to include protection studies in this very paper due to specificity of the animal models appropriate for SARS-Cov-2.

Regrettably, we are unable to expand the content of this article within the framework of this particular approach

 Why was the S protein immunization given subcutaneously rather that intramuscularly (which is more standard for immunization experiments) – describe what effect this might have had on speed and magnitude of systemic responses.

Reply. We thank the reviewer for this remark. The study evaluated the efficacy of sequentially administering vaccine antigens via oral and parenteral routes, with the subcutaneous route chosen as the parenteral method. Intramuscular injections offer a rapid delivery of the antigen into the bloodstream due to the abundant blood supply in muscle tissue. Subcutaneous administration results in a somewhat slower penetration of the antigen into the bloodstream, with absorption occurring through the lymphatic system before entering the bloodstream.

Despite variations in the speed of antigen delivery into the blood following injection, both routes effectively elicited rapid and robust systemic immune responses. In experiments involving mice, both subcutaneous and intramuscular routes of immunization are commonly employed. We opted for the subcutaneous method as it is considered less invasive compared to intramuscular injections.

I do not agree that some of their conclusions are valid when describing kinetics and magnitude of antibody responses especially in relation to L3 and L3-SARS. Need to re-evaluate which of these are correct with the evidence shown here and moderate these conclusions.

Reply. We thank the reviewer for this remark. In all our experiments assessing the kinetics and magnitude of the humoral and secretory immune response to the viral antigen in the probiotic strain E. faecium L3, biological materials from animals orally treated with unmodified E. faecium L3 served as a control. It was a conscious choice. This approach enables the monitoring of adaptive immune response development and allows conclusions to be drawn regarding its dynamics, considering only the excess of the studied indicators over nonspecific reactions in the control.

In the discussion section, we compared the nonspecific reactions of sera from control mice treated with unmodified E. faecium L3 with those from untreated mice in ELISA. Specifically, Supplementary Figure S5 illustrates that, in comparison to sera from untreated mice, those from mice treated with enterococci exhibit elevated levels of IgG capable of reacting with the S protein in the ELISA. Consequently, it can be assumed that if sera from untreated mice were utilized as a control to study the immune response to the probiotic vaccine L3-SARS, even more significant differences would be observed. These findings are interesting in the context of the relationships between the adaptive and nonspecific innate immune responses following the administration of a probiotic coronavirus vaccine.

 Avoid use of the term ‘live vaccine’ for the L3-SARS, whilst correct that it is a live organism, the term ‘live vaccine’ refers specifically to a live version of the target organism, so use a different term to describe the probiotic vaccine.

Reply. We agree that the term “live vaccine” puzzles many people doing classical vaccine studies (even though probiotic with vaccine antigen is live bacteria). In order to avoid misunderstanding we attempted to substitute the term "live vaccine" with the term "probiotic vaccine."

80-81: was it given 3 times with a two week period or 3 times with 2 weeks in between?

Reply. We thank the reviewer for this correction.  The vaccination protocol involved a single oral administration of the vaccine for three consecutive days, which was then repeated three times with two-week intervals. Thus, mice were orally administered the vaccine on days 1, 2, 3, 14, 15, 16, 28, 29, and 30 from the commencement of the experiment.

The text in the manuscript has been corrected.

Figure 1: was this from 3 consecutive days dosing, and does the x axis correspond to days from the first or the last dose?

 Reply. We thank the reviewer for this correction.  The X-axis corresponds to the days after the last dose. Figure 1 has been revised, and clarifications have been incorporated into the legend.

Figure 2: confusing. Please add detail to axes on figures so can understand what the measure is (even if can find it in the figure legend), e.g. mean serum IgG titer, mean serum IgA titer, mean nasal IgA titer.

Add detail to make clearer such as was provided in figs 3 and 5 with respect to serum treatment with L3, what the titer is referring to reciprocal log 10 or log 2 etc.

 Reply. We thank the reviewer for this correction.  The Figure 2 has been corrected.

Figure 3: what does ‘ads+’ mean?

Reply. We thank the reviewer for this correction.  It means “after adsorption”. We corrected Figure 3 and corrected “ads” to “adsorption”.

146: should read ‘mouse sera’, not ‘mice sera’

Reply. We thank the reviewer for this correction.  Corrected.

149: use 9-dose course (not 9-fold course)

Reply. We thank the reviewer for this correction.  Corrected.

Figure 4: add [IFN] pg/ml to y axis (easier to know what looking at)

Reply. We thank the reviewer for this correction.  Corrected.

175: consider using the term ‘magnitude’ not ‘strength’ as this could be misinterpreted as meaning the strength of binding of the antibodies (suggesting it was an avidity assay).

Reply. We thank the reviewer for this correction.  Corrected.

176: what do you mean by ‘intact mice’, (untreated/unvaccinated control mice?)

Reply. We thank the reviewer for this correction.  Corrected. We changed “intact mice” to “untreated”

Figure 5: add IgG or IgA titer in y axis title

Reply. We thank the reviewer for this correction.  Corrected.

Reviewer 2 Report

Comments and Suggestions for Authors

In this study, the authors development and assessment of a probiotic-based vaccine, L3-SARS, for COVID-19. The vaccine integrates a gene fragment encoding the spike protein of the SARS-CoV-2 virus into the probiotic strain E. faecium L3 and has been tested in mice for efficacy in inducing an immune response.

This research is valuable, but unfortunately there are only few experimental results. Therefore, the authors need to conduct more research on the safety and mechanism of immunity to reach the level of publication in the International Journal of Molecular Sciences.

 However, I have some suggestion,

1. The author mentions "Previous findings have demonstrated its efficacy in providing protection against the virus in Syrian hamsters." in the abstract, indicating that this research is a continuation of the research, and can briefly mention the results and provide references in the "Introduction".

2. The author mentions "Our findings indicate that the oral administration of L3-SARS elicits a modest adaptive immune response in mice" in the abstract. What does inducing a moderate "adaptive immune response" mean? It can be written clearly in the article.

3. Modified enterococci can still be isolated from feces 9 days after vaccination. But is the inserted gene fragment still in the modified enterococci?

4. Add titles to the figures in Figures 2, 3 and 5 to make them easier to read. For example,

Figure 2A: Sera IgG  2B: serum IgA  2C: nasal lavages IgA.

5. The Y-axis number in Figure 3 should be a comma (0.6), not a semicolon (0,6).

 6. Are lines 47 and 175 marked as errors?

 7. Do the mice that receive the vaccine have any health concerns? Such as changes in body weight, changes in the bacterial phase of the intestinal flora, or lesions in various organs?

 8. Since the coronavirus is constantly changing, is this vaccine also ineffective?

Author Response

We express our gratitude to all the reviewers for their thoughtful and constructive evaluation of our article. Their critical insights and kind considerations contribute significantly to enhancing the quality of our work and positively influence our understanding of the subject under study.

 Reviewer 2.

This research is valuable, but unfortunately there are only few experimental results. Therefore, the authors need to conduct more research on the safety and mechanism of immunity to reach the level of publication in the International Journal of Molecular Sciences.

 However, I have some suggestion,

 The author mentions "Previous findings have demonstrated its efficacy in providing protection against the virus in Syrian hamsters." in the abstract, indicating that this research is a continuation of the research, and can briefly mention the results and provide references in the "Introduction".

 Reply.         We thank the reviewer for this remark. The safety and protective efficacy of the current example of the recombinant probiotic vaccine against coronavirus infection were previously investigated in Syrian hamsters, and the findings have been published [Reference 26]. This information has been incorporated as an additional detail in the introduction section, and the corresponding reference number 26 has been included to cite the publication.

  1. The author mentions "Our findings indicate that the oral administration of L3-SARS elicits a modest adaptive immune response in mice" in the abstract. What does inducing a moderate "adaptive immune response" mean? It can be written clearly in the article.

 Reply.         We thank the reviewer for this remark. In the Abstract section, the evaluative term "modest" has been omitted. It is acknowledged that the immune response to orally administered antigens may not reach levels comparable to those achieved through parenteral introduction. The term "modest" was originally employed to convey this idea. Detailed discussions regarding the characteristics of the immune response are presented in the main body of the article.

 Modified enterococci can still be isolated from feces 9 days after vaccination. But is the inserted gene fragment still in the modified enterococci?

 Reply. We thank the reviewer for this remark. Indeed, up until day 9 we could detect modified enterococci. All the enterococcal colonies grown on erythromycin agar were analyzed by PCR. For amplification, we used a pair of primers flanking the inserted gene fragment on the side of the coronavirus part of the insert and the own DNA of the enterococcus adjacent to the insert. PCR results confirm the presence of such an insertion in colonies from the litter of vaccinated mice (Fig. S4). Resistance to erythromycin is a property of the vaccine strain of enterococcus. Enterococci, which belong to the natural intestinal flora of mice, cannot grow on media containing erythromycin. Supporting materials are provided in Supplementary Fig. S1-S3.

 Add titles to the figures in Figures 2, 3 and 5 to make them easier to read. For example,

Figure 2A: Sera IgG  2B: serum IgA  2C: nasal lavages IgA.

 Reply. We thank the reviewer for this correction. Corrected.

 The Y-axis number in Figure 3 should be a comma (0.6), not a semicolon (0,6).

 Reply. We thank the reviewer for this correction. Corrected.

 Are lines 47 and 175 marked as errors?

Reply. We thank the reviewer for this correction. We do not mark the above lines as errors.

 Do the mice that receive the vaccine have any health concerns? Such as changes in body weight, changes in the bacterial phase of the intestinal flora, or lesions in various organs?

 Reply. We thank the reviewer for this remark. No differences in the health status of mice were observed in the group receiving the vaccine strain compared to the control group receiving the original E. faecium L3 variant. The evaluation of the safety of the vaccine strain was not within the scope of this study, as it had been previously conducted. Safety assessments for this probiotic vaccine, incorporating an integrated DNA fragment of the coronavirus S protein, were performed and confirmed in Syrian hamsters, with the results published (Reference #26). While comprehensive safety evaluations from preclinical studies have been obtained, they are yet to be published. Our research team has been investigating the properties of the original probiotic strain E. faecium L3 for two decades, accumulating extensive evidence of its beneficial properties in animal experiments. This strain is utilized as a dietary supplement in humans.

 Since the coronavirus is constantly changing, is this vaccine also ineffective?

 Reply. We thank the reviewer for this remark. For the vaccine variant utilized in this study, the inserted DNA fragment encodes part of the S protein, specifically encompassing the RBD of the S protein of the Wuhan variant coronavirus. Consequently, the protective efficacy of the immune response to this coronavirus insertion is expected to depend to a lesser extent on the variable regions of the coronavirus genome in the RBD region, given the conservative nature of antigenic determinants within it.

         Previous studies with Syrian hamsters explored the protective efficacy of two vaccine strains of the probiotic vaccine—one with the aforementioned Wuhan variant and another with an inserted RBD fragment of the beta variant of coronavirus. When vaccinated animals were subsequently infected with the beta variant of SARS-CoV-2, the Wuhan vaccine strain demonstrated an 82% reduction in the replication of this heterologous virus in the lungs. Conversely, a homologous vaccine probiotic strain with a beta-S insert limited the proliferation of the beta variant of SARS-CoV-2 by 99%. Thus, the vaccine discussed in this article maintained a high degree of effectiveness against a later heterologous variant of coronavirus.

         It is evident that the creators of vaccines against the coronavirus face the challenge of keeping pace with its variability. The speed of preparing the current version of the vaccine becomes a fundamental question in this race. An alternative approach involves creating vaccines with broad specificity based on conserved antigenic structures of the coronavirus. The recombinant probiotic vaccines studied here may prove valuable for supporters of both approaches. Importantly, due to its ability to stimulate nonspecific innate immune defense mechanisms, this vaccine option is expected to have an inherent "initial" level of effectiveness owing to its probiotic potential.

         Thus, it can be hypothesized that this vaccine may be effective against current coronavirus variants. However, the extent of its effectiveness requires dedicated research to be determined.

Round 2

Reviewer 2 Report

Comments and Suggestions for Authors

Thanks to the author for the reply, but the research results are still too few to be published in the International Journal of Molecular Sciences.

Also, what does Figure 1A on line 173 mean?

Author Response

First and foremost, we want to thank the Reviewer and Editors for their careful consideration of our work.
Question:  "...what does Figure 1A on line 173 mean"

We especially thank the reviewer for this comment. We have made changes to the text to make our explanations as clear as possible.

Round 3

Reviewer 2 Report

Comments and Suggestions for Authors

no comment.